# Past and Current Progress in the Development of Antiviral/Antimicrobial Polymer Coating towards COVID-19 Prevention: A Review

**DOI:** 10.3390/polym13234234

**Published:** 2021-12-02

**Authors:** Nazihah Nasri, Arjulizan Rusli, Naozumi Teramoto, Mariatti Jaafar, Ku Marsilla Ku Ishak, Mohamad Danial Shafiq, Zuratul Ain Abdul Hamid

**Affiliations:** 1School of Materials & Mineral Resources Engineering, Universiti Sains Malaysia, Engineering Campus, Nibong Tebal 14300, Pulau Pinang, Malaysia; nasrinazihah@gmail.com (N.N.); arjulizan@usm.my (A.R.); mariatti@usm.my (M.J.); ku_marsilla@usm.my (K.M.K.I.); danialshafiq@usm.my (M.D.S.); 2Department of Applied Chemistry, Faculty of Engineering, Chiba Institute of Technology, 2-17-1 Tsudanuma, Narashino 275-0016, Chiba, Japan; teramoto.naozumi@it-chiba.ac.jp

**Keywords:** antimicrobial, antiviral, coating, COVID-19, nanoparticles, polymer coating properties

## Abstract

The astonishing outbreak of SARS-CoV-2 coronavirus, known as COVID-19, has attracted numerous research interests, particularly regarding fabricating antimicrobial surface coatings. This initiative is aimed at overcoming and minimizing viral and bacterial transmission to the human. When contaminated droplets from an infected individual land onto common surfaces, SARS-CoV-2 coronavirus is able to survive on various surfaces for up to 9 days. Thus, the possibility of virus transmission increases after touching or being in contact with contaminated surfaces. Herein, we aim to provide overviews of various types of antiviral and antimicrobial coating agents, such as antimicrobial polymer-based coating, metal-based coating, functional nanomaterial, and nanocomposite-based coating. The action mode for each type of antimicrobial agent against pathogens is elaborated. In addition, surface properties of the designed antiviral and antimicrobial polymer coating with their influencing factors are discussed in this review. This paper also exhibits several techniques on surface modification to improve surface properties. Various developed research on the development of antiviral/antimicrobial polymer coating to curb the COVID-19 pandemic are also presented in this review.

## 1. Introduction

Antimicrobial coatings generally refer to surfaces that contain antimicrobial agents that can inhibit the growth of microorganisms on surfaces. The incorporation of antimicrobial agents such as antibiotic compounds, quaternary ammonium compounds, or metal nanoparticles occurs through electrostatic or covalent interaction on surfaces, creating a surface-bound, active, antimicrobial, biocidal coating or a passive pathogen-repellent coating [1]. In the past, the development of an antimicrobial coating has been carried out extensively, such that this kind of coating is important and applicable in diverse applications. The usage of antifouling coating is part of the prevention of marine biofouling formation on the surfaces of marine structures such as boats, ship hulls, and pier columns [2,3]. Apart from the inhibition of microorganism growth, antimicrobial coatings have also been utilized for the shelf-life extension of fruits and vegetables either directly coated onto fruits and vegetables or as an active food packaging film [4,5,6,7,8]. In the medical and health fields, microbial contamination of medical devices may occur before or after implantation, which presents a significant challenge for the treatment of contaminated medical devices. Thus, the antimicrobial coating has been widely used in medical device manufacturing to prevent problems related to medical device-associated infections [9,10,11].

Recently, a terrific viral outbreak known as COVID-19 has caused more than a million global deaths and imposed severe morbidity on the human population regardless of age or gender. It was identified that COVID-19 is caused by a novel coronavirus SARS-CoV-2, which is closely related to SARS-CoV-1, and has become more virulent, with the number of infected cases increasing exponentially worldwide from day to day [12]. This outbreak has had negative impacts on human life and the global economy. Furthermore, the existence of new, fast-spreading variants of coronavirus SARS-CoV-2 worldwide has required a fast and effective solution to stop the viral spread [13]. Having multiple preventive interventions is important and, next to social distancing and maintaining high hygiene standards, several developments on new antiviral treatments and vaccines have been carried out to curb this outbreak. However, the processes involved in ensuring the good quality, effectiveness, and safety of vaccines and antiviral treatment before use are lengthy and complex. Indeed, the transmission of COVID-19 significantly occurs through direct person-to-person contact, and it is also recognized that there are possibilities of indirect transmission through contact with contaminated surfaces [14,15]. The contaminated droplets deriving from the cough droplets or sneezes of an infected individual will settle onto inanimate surfaces such as plastics, stainless steels, fabrics, and glass, where the coronavirus is able to survive for a specific period of time. Surprisingly, a review described that the survival time of coronavirus lasts up to 9 days on various objects at room temperature and low humidity [16]. A regular surface disinfection process could be performed using sodium hypochlorite (0.1%), hydrogen peroxide (0.5%), and ethanol (62–71%); however, the risk of virus transmission remains [17].

Concerning this problem, the generation of new, self-disinfecting surfaces or antimicrobial surface coatings may contribute to controlling the indirect transmission of COVID-19. In response to the disease transmission route, the purpose of antimicrobial surface coatings is to suppress the persistence of coronavirus SARS-CoV-2 on various surfaces that humans are exposed to every day [18]. Currently, most researchers have focused on developing antimicrobial coatings based on antimicrobial polymers and nanocomposites [19]. This is because polymer composites have been widely used for many objects and surfaces, such as medical devices in the healthcare sector, thus, an improvement in the antimicrobial activity of polymer composites against various pathogens and viruses could be useful in fighting against COVID-19. In addition, the usage of nanomaterial technology to protect against COVID-19 has been proven by the successful invention of a new vaccine based on lipid nanoparticle-formulated mRNA by Sahin and his colleagues [20]. Therefore, the incorporation of nanomaterials such as silver nanoparticles, as the antimicrobial agent, into a polymer can be a promising approach to developing new antimicrobial surface coatings with various antimicrobial performances. For instance, the addition of copper nanoparticles into a polymer matrix has high potential as an antiviral polymer coating that could be used for various surfaces [21].

A developed antimicrobial coating should be able to inactivate any virus and bacteria rapidly without leaving harmful effects for the consumers, and should be mechanically robust enough to coat any surface regardless of the applied environmental conditions [19]. Most of the previously reported studies focused on surface coating with antibacterial capabilities but unfortunately there has been less focus on antiviral surfaces and coatings [1]. Thus, the term antimicrobial will be used as the general term to characterize both coating surfaces with antibacterial and/or antiviral capabilities throughout this review. Several properties of coating surfaces for constructing antimicrobial coatings are reviewed in this paper. First, an overview on the classes of antimicrobial materials used for antimicrobial coatings are described. Several examples of antimicrobial agents or materials that have been used as a coating will be listed. Following this, some of the properties for an effective antiviral coating are elaborated into surface wettability, surface mechanism, and surface chemistry. This part includes some of the factors that influence surface properties. This review also explains several surface modification techniques in order to improve surface properties. Lastly, a few current studies and research on developing an antimicrobial coating against COVID-19 are also described in this review paper.

## 2. Classes of Antimicrobial Materials for Antimicrobial Coating

Antimicrobial materials or agents are promising candidates against various pathogenic microbes, drug-resistant bacteria, and viruses. Previously, the majority of microbial infections were treated by using antibiotic drugs. However, the usage of antibiotic drugs in developing an antimicrobial surface coating is not suitable, which may lead to antimicrobial resistance problems. For instance, the removal of biofilm formed on a surface through using antibiotics becomes difficult since the activity of antibiotics is limited at the top layer of biofilm and shields the bottom layer, resulting in developing antimicrobial resistance [22]. In addition, although antibiotics can save many lives, the increase in antibiotic application during the recent COVID-19 pandemic has become a concern as this may lead to the emergence of multi-drug resistant strains of infectious diseases [23]. Thus, an alternative to antibiotics is needed. Most current research studies are developing an antimicrobial coating based on various synthetic and natural antimicrobial agents or materials. The classification of antimicrobial agents or materials is shown in Figure 1.

### 2.1. Antimicrobial Polymers

An antimicrobial polymer is a polymer that has the ability to inhibit or inactivate the development or colonization of any microbes based on the presence of certain chemical structures on the polymer backbone or side chain, such as amphiphilic structures or polycation groups [24]. Antimicrobial polymers have non-specific mechanisms for inhibiting pathogen growth compared to discrete molecular antibiotics, such as bacterial lysis, via the disruption of the cell membrane due to different charges, while molecular antibiotics inhibit microbial DNA synthesis through receptor binding [25]. Thus, there is no development of microbial resistance against antimicrobial polymers. Currently, polymers are being used for antimicrobial surface coating fabrication due to their versatile macromolecular chemistry which assists in tailoring the physicochemical properties of polymers [26]. Antimicrobial polymers can be divided into several types and the choice of antimicrobial polymer group is dependent on the intended applications. Table 1 tabulates several examples of antimicrobial polymers with their antimicrobial activity in various applications.

The cationic polymer is the most used antimicrobial agent due to the presence of electropositively charged groups in the polymer chain. Poly(ethyleneimine) (PEI), poly [2-(*N*, *N*-dimethylamino) ethyl methacrylate] (PDMAEMA), and polydimethylsiloxanes (PDMS) are among the cationic polymers that are currently used as antimicrobial polymers. The biocidal action of cationic polymers occurs through a contact-dependent manner without any chemical release [44]. The cationic polymer is also able to resist the adhesion of bacteria and viruses on the surface [45]. The biocidal action of cationic groups starts with the adsorption of microbes onto the surface due to electrostatic interaction between positively charged groups of polymers and negatively charged groups of microbes, resulting in an increase in cell permeability and the disruption of cell membrane [46]. Clark and colleagues demonstrated that the adsorption of PEI in a negatively charged membrane could be confirmed by using X-ray reflectivity (XR) and Langmuir, which increases the integrity of the membrane under osmotic stress but causes overall deformation and leakage of the membrane [47]. It was previously demonstrated that an electrostatic interaction formed due to PEI adsorption triggered a substantial reorganization of the phospholipid membrane bilayer; lipid head groups were pulled inward to the membrane center causing transmembrane translocation of anionic lipids [48]. As a result, cell membrane damage will occur, leading to cell leakage (see Figure 2).

Other than synthetic cationic polymers such as PEI and PDMAEMA, a cationic polymer can also derive from natural sources such as chitosan, cellulose, dextran, and others [49]. These natural cationic polymers also have antimicrobial and biocidal properties similar to the synthetic polymer. Interestingly, natural cationic polymers are biodegradable, non-toxic, and biocompatible, and their properties can be improved via modification at the reactive site [50]. Among the mentioned natural polymers, chitosan is commonly used for the development of an antimicrobial coating for biomedical devices, drugs, food, and others [40,51,52,53]. Chitosan has positively charged, functional amino groups of the chitosan chain, and is insoluble in most solvents at a neutral or high pH value above 6.5. To overcome this limitation, chitosan derivatives, such as *N*,*N*,*N*-trimethylchitosan (TMC) and O-imidazolyl quaternary ammonium chitosan, have been synthesized via chemical modifications to improve water solubility of chitosan. The introduction of a quaternary ammonium salt group inside chitosan can be performed via several methods such as direct quaternary ammonium substitution, N-alkylation, and the epoxy-derivative open loop method.

Direct quaternization or alkylation of chitosan has been carried out by using methylating agent (dimethylsulfate or methyl iodide) in the presence of a strong base (sodium hydroxide, NaOH) and catalyst, producing TMC [54,55,56]. Sometimes, aqueous acid solutions such as ascorbate and citrate are used to replace iodide ions in trimethylchitosan iodide that are not safe in some applications. Besides, glycidyl trimethyl ammonium chloride (GTA) or (3-chloro-2-hydroxypropyl) trimethyl ammonium chloride (CTA) are also quaternary ammonium salts that have been used to modify chitosan through an epoxy-derivative ring-opening process. The reaction occurs at C_2_–NH_2_ in chitosan under alkaline conditions. Several chitosan quaternization reactions are shown in Figure 3. Furthermore, a quaternized chitosan has shown strong antimicrobial effects against *Staphylococcus aureus* and *Escherichia coli* because of having higher polymeric charge density compared to commercial chitosan [56]. The presence of a new quaternary ammonium salt group into commercial chitosan not only improved water solubility but also increased chargeability, which subsequently enhancing antibacterial activity. Interestingly, TMC, the quaternized chitosan, is able to form complexes with bacterial cell membranes and interrupt gene expression activity within bacteria [57].

Furthermore, the usage of an amphiphilic polymer is effective as an antiviral coating agent because of the presence of hydrophobic and hydrophilic components in the polymer structure. A review suggested that an ideal amphiphilic antimicrobial polymer bearing a cationic arm has low molecular weight, a low level of lipophilicity, and is biocompatible towards eukaryotic cells [24]. An amphiphilic copolymer, P(DMAEMA-co-MMA), as shown in Figure 4, was successfully synthesized through the free radical polymerization of various concentrations of hydrophilic monomer 2-dimethylamino ethylmethacrylate (DMAEMA) and hydrophobic monomer methyl methacrylate (MMA) [58]. It was found that the copolymer with a high concentration of DMAEMA showed high biocidal activity against Gram-positive bacteria due to the presence of more amine groups along the DMAEMA chain.

The antimicrobial activity of an amphiphilic polymer such as poly (ester urethane) has also been influenced by the presence of a hydrophobic group wherein amphiphilic polymers with more hydrophobic pendant groups demonstrated strong antimicrobial effects against both Gram-positive and Gram-negative microbes [59]. Unfortunately, the toxicity of the polymer against mammalian cells also increased as the hydrophobic groups increased, which is also consistent with the observation from a study by Cuervo-Rodríguez et al. [60]. Since the cellular surface of eukaryotic cells lack net negative charge and are made up of a zwitterionic lipid head group with a cholesterol component, hydrophobic interaction is aroused between the hydrophobic group in the polymer chain and the lipid bilayer of red blood cells. Thus, hydrophobic interaction could induce hemolysis capacity [61]. Therefore, amphiphilic balance (hydrophilic/hydrophobic) must be considered during the designing of an amphiphilic polymer with low hemolysis activity such as a polymer bearing a high alkyl chain with ester or amide moieties [62].

Surfactant is one of the amphiphilic molecules and is classified as amphoteric, cationic, anionic, or non-ionic based on the charge present on the polar head. Surfactant has showed its potential as an antimicrobial agent against several types of microbes. In one case, cationic micelle surfactant with amide moieties was able to disrupt the membrane integrity of *Escherichia coli* and was slightly toxic towards mammalian cells [63]. Furthermore, surfactant embedded inside a polymer can be used for an antimicrobial surface coating. A multilayer film made up of a complex of modified poly(acrylic) acid with cationic surfactants was developed and portrayed good antimicrobial properties against numerous pathogens [64]. The amphiphilic structure of surfactant is useful for antiviral activity. The fatty acid chain of surfactant attaches to the hydrophobic component found on the virus, causing virus defragmentation, as shown in Figure 5. Thereafter, the virus fragments will be entrapped inside micelles and washed away due to how the hydrophilic head of surfactant develops a significant interaction with water that can be easily washed off from the surface [65]. In addition, surfactant can also act as a viral inhibitor as it will bind to the protein receptor on a virus surface such as influenza hemagglutinin [66].

A polymer containing a halogen element such as fluorine, bromine, chlorine, and iodine can also be a potential antimicrobial agent. Usually, a halogen-containing solution such as bleach is used as a disinfecting solution applied to household cleaning products. The presence of chlorine in the form of hypochlorite in bleach as the active agent makes it safe to be used by humans. Due to their unique properties, such as strong electronegativity and water/oil repellency, the oxidative agent, halogen, has a broad spectrum of antimicrobial and antiviral activity via antibiofilm formation and contact-killing action [26]. Based on molecular docking and molecular dynamic simulation, halogenated dopamine methacrylamide (DMA) was able to inhibit bacterial fatty acid synthesis due to hydrophobic interaction and van der Waals forces with an enzyme binary complex [67]. In addition, the position of a halogen substituent in a compound will also influence antimicrobial and antiviral activity [68,69,70]. A study by Stefanska et al. (2015) found that halogen at the third position of a phenyl group is significantly important for antimicrobial activity because of the increase in electronegativity [69].

*N*-halamine compounds are defined as compounds containing one or more nitrogen–halogen covalent bond as a result of a halogenation process on imide, amide, or imine groups [71]. This covalent binding is designed to provide stability, and will eventually slowly release free halogens into the environment or directly transfer them into the active site of a biological target site [26]. *N*-halamine polymer is synthesized by covalent bonding between an *N*-halamine precursor and a targeted polymer. As an example, Kocer et al. (2011) synthesized a copolymer of 3-chloro-2-hydroxypropylmethacrylate and glycidyl methacrylate coated on a cotton substrate and treated with 5,5-dimethylhydantoin, an *N*-halamine precursor, which provided antimicrobial properties [72]. The polymer coating successfully inactivated Gram-positive and Gram-negative bacteria within minutes of exposure. In addition, wound dressing containing an *N*-halamine compound showed no cytotoxicity against cell viability while preventing microbial infection compared to commercial wound dressing [73].

### 2.2. Metal Ions/Metal Oxide-Based Antimicrobial Agents

Metal ions and metal oxides are inorganic materials that have been investigated for their use as antimicrobial and antiviral agents in various fields, including medicine, food storage and preservation, and water treatment. Silver ions, copper ions, zinc oxide, and other metal-based antimicrobial agents have a potent toxicity effect against numerous pathogens at an exceptionally low concentration [74]. These metals ions or oxides share similar antimicrobial and antiviral modes of action over different microbe strains and species. The possible mode of action can be a metal reduction potential or metal donor atom selectivity/speciation [74].

The metal reduction potential mechanism is involved in a redox reaction that determines the tendency of a metal to acquire electrons from a donor, thus reducing the metal. With the presence of metals, reactive oxygen species (ROS) such as hydroxyl radicals (OH^−^), superoxide (O_2_^−^), and hydrogen peroxide (H_2_O_2_) are the products of redox reaction and are highly reactive molecules. These ROS species induce oxidative stress inside the cell. For example, Fenton reaction is one of the redox reactions that occurs in a cell wherein iron metal in reduced form reacts with H_2_O_2_, producing more reactive oxygen radicals, as shown in the following Equation (1):
(1)$${\mathrm{Fe}}^{2+}+H_{2\;}O_{2}\;\to \;{\mathrm{Fe}}^{3+}+{\;\mathrm{OH}}^{-}+\mathrm{OH}\cdot$$

In a study by Ali et al., the uptake of zinc ions by *Escherichia coli*, *Pseudomonas aeruginosa*, and *Staphylococcus aureus* showed a significant increase in ROS production [75]. These ROS species are identified as unstable molecules that tend to disrupt other components found in bacterial cells and viruses. Warnes and his research group demonstrated viral genome destruction and damage to viral capsid, leading to murine norovirus inactivation once exposed to a copper-based surface [76]. This was then supported by a following study that proved the rapid inactivation of human coronavirus 229E on a copper alloy surface was enhanced due to the generation of ROS species on the surface [77]. Apart from direct ROS production, metal poisoning also causes the oxidation of cellular thiols by forming covalent bonds between metal atoms and S (sulfur). Thiol groups are usually found in essential pathway such as cell wall formation cellular respiratory system [78]. As a result, there is a formation of protein disulfide and a reduction in antioxidants such as glutathione within the cell microbes [74]. This indirectly increases the generation of ROS species inside the microbial cells. 

Metal donor atom selectivity is another possible mechanism for antimicrobial activity. A unique property of a metal atom or ion is the ability to selectively bind to any donor atom, forming a complex with the donor groups such as oxygen, nitrogen, and sulfur. Therefore, the presence of metal atoms or ions inside the microbial cell starts to displace any original metal present inside the cell microbe. For instance, copper ions have proven to displace iron atoms from dehydratase iron-sulfur clusters, the enzyme involved in branched-chain amino acid biosynthetic pathway [79]. Thus, there is now damage to organelles such as nucleic acid, and protein membranes occur, leading to oxidative stress [80,81]. The antimicrobial mechanism of copper ions is illustrated in Figure 6.

There are a lot of metal ions and metal oxides, such as copper, silver, zinc, and titanium oxide, which are promising candidates that show effective antiviral activity for various purposes such as surface coating, a biosensor, cancer treatment, and drug development [82,83,84,85]. Several studies on the effectiveness of the antiviral activity of metal against various types of virus have been reported such as on murine norovirus, human coronavirus 229E (HuCov-229E), H1N1 influenza A, human immunodeficiency virus (HIV), and herpes simplex virus (HSV) [76,77,86,87]. Silver metal has been used as an effective antimicrobial and antiviral material. This metal has the ability to inactivate various viruses by interacting with the viral envelop and surface protein, inhibiting viral entry into cells, interrupting essential pathways such as the respiratory pathway, binding onto viral genomes, and interacting with viral replication factors [1,88,89]. Meanwhile, copper metal as an antiviral agent inhibits the growth of a virus by targeting the viral genome as the main antiviral mechanism mentioned in a study by Warnes’ group [77]. Zinc metal has also shown its antiviral activity by inhibiting the proteolytic cleavage, resulting in halting the synthesis of viral polypeptide, interfering in viral replication through free virus inactivation together with the inhibition of viral uncoating, viral genome transcription, viral protein translation, and polyprotein processing [90]. Interestingly, titanium dioxide, which is known as a photocatalyst, can also be a potential antiviral agent owing to its properties. As an example, under the exposure of UV irradiation for an hour, strong antiviral activity was observed at a low concentration of titanium dioxide-modified hydroxyapatite composite with the highest reducing titer rate, approximately more than 2 log/h [91]. 

As a strategy to enhance antiviral activity, metals in the form of ions, oxides, or nanoparticles are incorporated into a polymer matrix or with other antimicrobial metals to form a composite material. A study showed that the incorporation of silver nanoparticles into a chitosan matrix depicted antiviral activity in comparison with neat chitosan [92]. Here, the chitosan matrix created spatial restriction for the interaction of silver nanoparticles with the virus, thus preventing the physical binding of the virus on host cells. A study by Park et al., proved that silver nanoparticles decorated with silica coating can also inhibit the growth of influenza A virus [93]. Antiviral activity is involved in the generation of ROS by silver nanoparticles and silver ions and in the inhibition of hemagglutinin and neuraminidase activities on viral surface.

Moreover, a polylactide (PLA) film with silver nanoparticle impregnation was successfully fabricated for application in food packaging by using a solvent casting technique [4]. From the in vitro results, the evaluation of the antibacterial and antiviral activity of this film against Salmonella and feline calicivirus showed an increased as the concentration of impregnated silver increased. Furthermore, the PLA-silver nanoparticle film still retains its antibacterial and antiviral activity after five washings. The sustained release mechanism of silver ions from the PLA film may contribute to maintaining an antimicrobial effect during storage time and after the washing process.

In another study, a hybrid coating consisting of various cations such as copper, silver, and zinc prepared by radical polymerization via a sol–gel method demonstrated a high titer reduction in various viruses [94]. The hybrid coating on glass slides did not show a significant decrease in metal ion content within the hybrid after four hours’ span. This observation concluded that there were metal cations in leachate from the coating showing a virucidal effect by direct adsorption onto a viral membrane. However, this hybrid coating was not effective against non-enveloped viruses such as rhinovirus due to the lack of a lipid bilayer envelope to provide a virally encoded receptor-binding protein for virucidal targeting. Recently, a reformulation of the ethanol–zinc formulation was performed by the addition of nickel chloride salt and was tested against non-enveloped virus [95]. These divalent ions and salt induced capsid structural change and a reduction in capsid rigidity, which in turn has the potential to be used as active ingredients in disinfectant.

### 2.3. Antimicrobial Nanomaterials

The application of nanotechnology as part of a strategy for developing antimicrobial materials is currently being explored and more research is focused on this technology. Nanomaterials are materials with a size range within 1 to 100 nm that have enormous ability to inactivate or kill microbes, including viruses [96]. Due to their physical form, involving a large surface area and various shapes, they possess numerous unique properties, such as the ability to impart different colors due to their absorption in a visible region, and their reactivity and toughness [97]. They have been used in various fields such as drug delivery systems, bacterial and viral detection assays or biosensors, and disease treatment [98,99,100,101,102,103]. In addition, these nanomaterials, especially nanoparticles, have a broad spectrum of antimicrobial activity against various pathogens and viruses compared to any other antimicrobial agents [104,105]. The usage of nanomaterials is usually incorporated into various polymer matrices to provide a large surface area compared to bulk materials [74,106]. Therefore, the application of nanomaterials is feasible for effective antimicrobial surface coating. 

There are several physicochemical properties of nanoparticles influencing the antimicrobial activities of nanoparticles, such as nanoparticle size, surface chemistry, shape, size distribution, morphology, agglomeration, and dissolution rate [107]. These physicochemical properties of nanoparticles can be controlled by regulating the conditions during the nanoparticle synthesis process, and several examples are described in Table 2 [108,109,110]. A size dependency of embedded silver nanoparticles (AgNPs) on antiviral activity against H1N1 influenza A virus had been demonstrated wherein a smaller size of AgNP showed stronger antiviral activity [92]. Consistently, similar result also showed that AgNPs with a diameter smaller than 10 nm exhibited strong inhibition activity against SARS-CoV-2 [111]. It can be assumed that smaller nanoparticles have high-stability binding interaction with available protein on viruses. It has been stated that nanoparticles tend to be attracted to sulfur-bearing residues of glycoprotein knobs on viral membrane [112]. In addition, the size of the virus is around 20 nm to 1000 nm in diameter, which is easier for the penetration of nanoparticles into virus. 

The surface chemistry of nanoparticles is related to the surface charge available on their surfaces, which is provided by capping agents. A previous comparative study showed the toxicity effect on *Escherichia coli* between positively and negatively charged AgNPs [129]. The results showed that positively charged AgNPs were more toxic than negatively charged particles due to electrostatic interaction with negatively charged bacterial membranes. The antimicrobial activity of negatively charged nanoparticles may only be observed at high concentrations due to molecular crowding which leads to a net interaction of nanoparticles with microbial cells [130]. Hence, positively charged nanoparticles are most effective as antiviral agents since viral membrane is also negatively charged. Nanoparticles are also available in various shapes, such as spherical, triangular, and plane shape, which influences antimicrobial activity. A study conducted by Cheon et al., stated that different shapes of AgNPs showed different antimicrobial activities and is associated with the ion release rate from nanoparticles due to the difference surface areas of AgNPs [131]. Spherical nanoparticles may confer the highest reactivity and greatest stability for the binding process of microbial cells. Some studies concluded that spherical-shaped nanoparticles have higher antimicrobial activity than others [92,123,132]. In contrast, a zinc oxide (ZnO) nanopyramid depicted the highest biomimetic enzyme inhibition as it provides a geometrical match with the enzyme [133]. This anisotropic shape has greater antimicrobial activity because of its basal plane with a high atom density (111); these are facets that provide maximum reactivity at site [134].

Metal-based nanoparticles are the most frequently used inorganic nanoparticles and are used as antimicrobial agents such as silver (Ag), copper (Cu), zinc (Zn), gold (Au), and many more. The antimicrobial activities of inorganic metal nanoparticles involve the generation of ROS, the physical disruption of cell membrane permeability and integrity, induced changes in protein structure, and the inhibition of DNA and RNA replication [135]. The antimicrobial mechanisms for metals in the form of nanoparticles, ions, or in bulk form are quite similar, but there are differences in the strength of their antimicrobial effect. Wang’s group identified that the toxicity of metal oxide nanoparticles could originate from metal ions released, from nanoparticles themselves, and from both ions and nanoparticles that interact with pathogens and viruses [136]. As illustrated in Figure 7, the antiviral mechanism of metal nanoparticles involved in the interaction with virus particles includes binding with cellular factors and viral factor, which will further block viral replication. Hence, the usage of metal nanoparticles in surface coating fabrication is better than other forms due to its multivalent methods for microbial inhibition. For instance, an antiviral composite coating made up of a silver nanocluster embedded in a silica matrix was successfully deposited on disposable a face mask via sputtering process [128].

Carbon-based nanomaterial is another category of nanomaterial aside from metal based. Carbon-based nanomaterials have become a promising antiviral agent with a broad spectrum of enveloped viruses and no toxicity towards humans [137,138]. Carbon dots (CDots) are one of the family members of carbon-based nanomaterials. They have unique properties such as high surface to volume ratio, they contain the chemical functionalities of organic molecules, and they can be homogenously dispersed in water. As an antiviral agent, functionalized CDots from 2,2′-(ethylenedioxy)bis(ethylamine), (EDA)-CDots, and 3-ethoxypropylamine (EPA)-CDots were found to be effective for inactivating human norovirus virus-like particles [139]. No effect was observed on the integrity of viral capsid protein and viral particles, but the inhibition of a virus binding onto the receptors of human cells was observed. This study also concluded that the surface charge available on CDots influences their antiviral activity. Besides CDots, fullerene is another nanocarbon family that is hydrophobic in nature and has also been used as an antiviral agent. A derivative of fullerene has shown a strong inhibition of HIV-1 maturation [140]. This derivative compound impaired viral polyprotein processing through a protease-independent mechanism.

## 3. Surface Properties for Antimicrobial Surface Coating

In order to produce the most reliable and efficient antiviral surface coating, several polymers’ surface properties, such as their wettability, mechanical stability, and surface chemistry, should be considered. Surface properties of a coating are dependent on each other. For instance, information obtained from the wettability of a surface can be used to estimate the polarity of the surface, including the chemical composition and functional groups found on the coating surface. Hence, several surface properties are elaborated below to produce an ideal surface coating.

### 3.1. Surface Wettability Properties

The wettability of a material surface is important for protection purposes, metal anticorrosion purposes, and for a self-cleaning surface [141,142,143]. Regarding antimicrobial or antiviral function, the wettability of an antimicrobial or antiviral surface may affect the adsorption of microbial cells, the formation of biofilm, and the survival of pathogens [144]. Surface wettability is actually related to the interaction of water on the surface and it is determined by using a contact angle (θ), which is defined as the angle between the liquid–air interface and the solid surface [145]. When a liquid droplet is deposited on a solid plane surface, the droplet forms a shape with a certain contact angle to the solid surface. Wetting on a surface plane involves an equilibrium contact angle, which is a characteristic measure of the energy state between three interfacial surface boundaries [146]. These are the solid–air interface (sv), solid–liquid interface (sl), and liquid–air interface (lv). Thus, the final value of the contact angle is measured when it satisfies the equilibrium state of these interfacial boundaries. The contact line of three different interfacial boundaries is shown in Figure 8.

Young suggested that wettability involves an interaction between the forces of adhesion between liquid and solid surface and the forces of cohesion within the liquid molecules; these determine the occurrence of wetting on a solid surface [147]. Young’s equation, as shown in Equation (2), describes the balance between three interfacial phases:
(2)$$\gamma_{\mathrm{lv}}\;\cos\theta ={\;\gamma }_{\mathrm{sv}}-{\;\gamma }_{\mathrm{sl}}$$
where γ denotes the interfacial tensions dependent on the phases and θ is the contact angle between the tangent of liquid–air interfacial and solid surface. Based on the Young equation, a hydrophilic surface usually has a high surface energy in order to attract the molecules of a liquid to the surface, resulting in the spreading of the droplet. In a simpler way, the surface energy (adhesive force) is stronger than the surface tension of liquid (cohesive force), which is normally constant, thus causing the liquid to spread over the surface and creating a low contact angle. A hydrophilic surface usually has a contact angle lower than 90°. However, a hydrophobic surface has a lower surface energy than the surface tension of liquid, producing a bead-shape liquid droplet with a contact angle greater than 90°. Figure 9 shows the difference in contact angle between a hydrophilic surface and a hydrophobic surface.

The wettability of a surface is influenced by surface roughness, and the effect of roughness on surface wettability is further explained based on the Wenzel model and the Cassie–Baxter model. The Wenzel model suggests that as liquid droplet will completely spread over the surface, including within the roughness structures (Figure 10a), thus the apparent contact angle is described as the following Equation (3):
(3)$$\cos_{A}=R\;\cos\theta$$
where θ_A_ is the apparent contact angle and R is the ratio of solid–liquid contact area to the projected area of the solid surface for which the value of R is bigger than 1. Therefore, when the rough surface is fully wetted by liquid droplets, surface roughness intensifies the wetting properties, and the R value remains bigger than 1. Meanwhile, the Cassie–Baxter model introduced the idea that a liquid droplet can be supported on top of a roughness structure, leaving trapped air in between roughness structures (Figure 10b). The apparent contact angle can be calculated based on the following Equation (4):
(4)$$\cos\theta_{A}=\mathrm{Rf}\;\cos\theta +f-1$$
where f is the fraction of solid area to solid–liquid area.

Surface roughness has resulted in a variability in surface wettability behavior, subsequently influencing the antimicrobial activity of a surface. The presence of porous structures in a coating surface will affect the roughness and wetting properties of a surface coating. In a study by Almasi and colleagues, a newly developed antimicrobial microemulsion film with high internal porosity and surface roughness made that film become hydrophilic over time [148]. This was because of the presence of microemulsion ingredients such as oil, surfactant, and co-surfactant inside the polymeric network structure producing more porous fibrous structures. The high roughness value (R_a_) was attributed to the presence of colloidal particles on the surface. Thus, the wettability of the film increased.

Similar to findings from Hosseini’s group, a porous hydrophilic cupric oxide coating film showed an excellent SARS-CoV-2 reduction of 99.8% within 30 min of exposure [149]. The hydrophilicity of this coating had enhanced the contact area between viral suspension and cupric oxide coating by the rapid infiltration of viral suspension into the porous structures. Therefore, a shorter viral inactivation time via a hydrophilic coating could minimize the probability of viral infection for the next user of contaminated surfaces.

The roughness of a surface can also be due to the presence of micro/nanostructures on a surface coating, which in turn can affect the formation of biofilm. Lutey and co-workers demonstrated and compared the effect of different laser-textured surfaces on surface wettability and bacterial retention for antimicrobial purposes [150]. Different surface morphologies were produced by varying several parameters during laser treatment. It was observed that laser treated surfaces with larger surface features had high wettability, which influenced the attachment or retention of bacteria onto the surface. On the contrary, the other two laser treated surfaces with fine surface features had a high water contact angle that successfully reduced bacterial retention. These fine surface features decreased the contact area between bacterial cell and contact surface.

Siddiquie’s study fabricated an antiviral hydrophobic surface by creating micro/nanopillar structures on the plane surface via a femtosecond laser process, resulting in an antibiofouling effect [151]. These pillar structures reduced the chance for virus adsorption onto the surface because of a smaller contact area between virus and surface. According to the Cassie–Baxter model, the presence of entrapped air in-between the nanopillar structures was attributed to a decrease in the water contact angle following the reduction in bacterial adhesion.

Next, a hydrophobic antiviral surface coating with a high contact angle, approximately 130°, was successfully fabricated by embedding cuprous oxide particles into commercialized polyurethane film [152]. The hydrophobic surface coating treated with SARS-CoV-2 was found to have a small amount of the virus on the surface after an hour of treatment. The research group added that the wettability of an antimicrobial coating may be attributed to the time course of inactivation of SARS-CoV-2; a hydrophobic coating required a longer time for the contaminated droplets to reach or penetrate through the antimicrobial coating. Meanwhile, surface coating with a low water contact angle was able to reduce the virus particles by 99.9% after one hour. Furthermore, the level of hydrophobicity of an antimicrobial surface can be influenced by the concentration of additional antimicrobial agents incorporated into a polymer coating base. Sarimai’s published research revealed that an increase in copper oxide nanoparticle concentration results in an increase in surface coating hydrophobicity [153]. However, the water contact angle of the surface coating dropped after reaching a certain concentration of copper oxide nanoparticles, which indicated that the stability between copper oxide and polystyrene (hydrophobic coating) was limited.

Moreover, a superhydrophobic surface is a surface which has a contact angle of more than 150° which is useful for inhibiting the adsorption of microbes onto a surface. This surface is known to have self-cleaning properties that make it easy for disinfection processes. Recently, a plasmonic and superhydrophobic N95 respirator was developed by applying polyacrylamide film with deposited silver nanoparticles and graphene via pulse-mode, laser-induced forward transfer treatment onto the N95 respirator [154]. The presence of laser-induced graphene layers provided a superhydrophobic surface and thus helped to roll off any respiration droplets that may have been contaminated with viruses and pathogens from the respirator surface.

Another attempt at superhydrophobic self-cleaning surface fabrication was made by Milionis’ group [155]. They introduced two different hydrophobic layers, water-soluble fluoroalkylsilane (FAS) and ethanol-soluble stearic acid (SA), over zinc oxide nanostructure surfaces. After hydrophobization treatments, the measured water contact angle for FAS and SA layers were 158° and 160°, respectively. As a result, this hydrophobization process successfully enhanced the antimicrobial activity of the zinc oxide nanostructures without interrupting the surface geometry. Hence, this feature can be suitable and applicable for any surfaces that are vulnerable to a viral threat.

Note that most of the hydrophobic surface coatings were able to repel or reduce bacterial adhesion as part of their antimicrobial mechanism, while hydrophilic surfaces tend to promote bacterial attachment. However, some studies showed contrary results. A study by Wassmann and colleagues found that more bacterial adhesion was observed on hydrophobic surfaces than hydrophilic surfaces for *Staphylococcus epidermidis* bacteria [156]. This could be due to the hydrophobic properties of the bacteria that could be explained by the thermodynamic model of microbial adhesion. Yuan and co-workers demonstrated that a superhydrophilic plasma-treated surface with negative zeta potential was successful at limiting bacterial binding due two possible reasons [157]; these being a repulsive force arising between a negatively charged bacterial cell wall and a negatively charged surface, and the reduction in hydrophobic interaction. This result corresponds similarly with a previous study by Yoon’s group [158]. The superhydrophilic nanocomposite coating showed low bacterial adhesion that could also be due to the formation of a hydration layer on the surface that acts as ‘water shield’, thus restricting electrostatic interaction.

Overall, surface wettability properties play an important role in antibacterial adhesion on a surface. These wetting properties can be tuned by controlling the surface roughness and surface topography, including the additional materials used during surface coating preparation. Nevertheless, the adsorption of bacteria on a surface could be dependent on the type of bacteria itself and the interaction between bacteria and a surface regardless of surface wettability.

### 3.2. Surface Mechanical Properties

The mechanical properties of a surface coating involve the evaluation of robustness, the durability or long-lasting stability of a coating over a period of time and under extreme conditions, such as UV irradiation degradation, acidic, or alkaline conditions, and mechanical stress. One of the problems in commercializing these surface coating products is their ability to maintain their unique function and stability after several usages [19]. There are components present inside polymer coatings that may degrade once expose to unfavorable conditions. This can cause surface deformation and lead to changes in surface and morphological properties, thus resulting in the loss of their function [159]. A surface coating which has weak adhesion strength may also lead to the delamination of coating from a substrate surface [160]. Hence, the best surface coating should have strong bonding between coating and substrate to withstand harsh conditions.

As for antimicrobial and antiviral purposes, it is important for this coating film to have effective hardness, flexibility, wear resistance, and adhesion strength onto polymer substrates such as biomedical devices, implants, or common surfaces of objects including handrails and doorknobs. However, there is not so much research on the development of antimicrobial and antiviral coating surfaces with enhanced mechanical strength. Thus, some studies on good mechanical coatings can be employed or referred to when making a coating with good antimicrobial properties and mechanical strength.

Surface modification treatment is one of the methods used to improve the mechanical performance of coating onto a polymer substrate. This is because of structural surface changes, such as surface roughness and porosity, which may affect the performance of coating strength. There are several surface modification methods employed in order to deposit a coating with strong adhesion onto polymer substrates. The effect of different types of surface treatments on polymeric surfaces such as mechanical (sand grit blasting), chemical (primer coating), and energetic treatment (UV/ozone radiation) on bonding strength were compared in a previous study [161]. The results showed that different surface treatments produced a variability in adhesion strength based on different polymeric systems and adhesives used. For instance, an ethylene propylene diene methylene (EPDM) surface did not show any improvement in adhesive strength, while all treatments were effective when a silicon-based adhesive was used on a polyvinyl chloride (PVC) surface. This study can be extended to explore the effect of each surface treatment on antimicrobial properties while maintaining good adhesion strength.

Abrasive blasting process is a mechanical technique used to change the properties of surfaces, such as roughness and wettability. Rocha’s group successfully developed a composite coating made up of hydroxyapatite and titanium dioxide (HAp-TiO_2_) employed on a titanium alloy implant via plasma thermal spraying [162]. Substrate roughness was improved by using abrasive alumina blasting treatment prior to the coating process. Improving surface roughness helps in the mechanical anchoring of HAp-TiO_2_ coating. In addition, heat treatment resulted in microstructures of HAp-TiO_2_ being produced and arranged in successive layers to form a coating. As a result, the adhesion strength of composite coating increased to 30 ± 2 MPa. In previous study, the influence of size of grit particles (50 µm and 250–320 µm) used in sand blasting on various surface properties was investigated [163]. The pretreatment caused surface morphological changes, pits formation, and caused it to become rougher as the grit size increased. Following the pretreatment was an acid etching process that slightly reduced the surface roughness. However, based on the results of scratch test, it was found that the pretreatment of a titanium surface with corundum powder with a size of 50 µm was more favorable than other particle sizes as it showed higher resistance against plastic deformation than the other sample.

Chemical surface modification can also be performed and is better than a mechanical technique because it provides strong chemical bonding that cannot be easily interrupted. For instance, an alkali activation surface involves introducing a new functional group at the active site for further anchorage. Reggente’s research group created a novel alkali surface activation technique to modify the low surface energy of titanium substrate to enhance the adhesion strength between titanium substrate and poly (methyl methacrylate) (PMMA) polymer chain [164]. The surface activation of titanium substrate was carried out in sodium hydroxide solution with new hydroxyl groups attached onto the surface. These hydroxyl active sites provided an anchorage site for phosphonic acid, a coupling agent and an initiator for atom transfer radical polymerization (ATRP) of PMMA on the substrate. Based on morphological observation, there was a formation of a hierarchical porous interlayer along with a pillared and cone-shaped structure on the surface layer. Similar observations had been reported in previous studies [165,166]. These porous structures are useful for mechanical interlocking between substrate and PMMA-grafted layer, thus improving the adhesion force of coating by approximately 260 MPa (the critical force is 3.5 N).

Apart from various surface modification techniques, the molecular composition present inside polymer coatings is one of the factors affecting mechanical strength [167]. Through thermal cross-linking of polymer coating, a polyvinyl pyrrolidone (PVP) chain, stabilized with a polyethyleneglycoldiacrylate (PEGDA) cross-linked matrix onto a polymer substrate, polypropylene (PP) was developed with enhanced adhesion strength [10]. A different blend ratio of PEGDA–PVP played an important role in influencing the adhesion strength of a coating; as the proportion of PEGDA increased, the adhesion strength and flexibility of coating also increased, while coating hardness decreased until it reached maximum strength value when PEGDA was in excess. This is because the addition of PEDGA in the blend ratio will form further molecular entanglement between PEGDA and PP surfaces.

The latest study also evaluated the effect of silver content in composite film on the mechanical and structural properties of the thin films [168]. The film was deposited on a silicon substrate via physical vapor deposition–magnetron sputtering where various silver content ranged from 10 to 35%. It was found that there was a transition from an amorphous structure to a crystalline structure for silver content above 25%. The changes in structural form influence mechanical properties such as hardness and adhesion strength, which increase and decrease, respectively, when the silver content is above 25%. It was proven that the addition of any filler material, such as silver nanoparticles, could enhance mechanical strength [169].

Furthermore, the design of a surface coating may influence surface wear resistance as part of mechanical stability. A coating with a ‘sandwich’ structure, such as the coating composed of a chemically etched aluminum alloy surface with a silicon dioxide hybridized silane layer in the middle, followed by a carbide nanosheet hybridized silane layer on the top surface, could contribute towards greater wear resistance [170]. This type of coating design was able to maintain its superhydrophobic feature after 20 cycles of abrasion testing and only slightly reduced after 30 cycles of abrasion testing, thus showing greater wear resistance.

Interestingly, bacterial adhesion and biofilm formation are influenced by the stiffness of a surface coating material. There were few studies published on the effect of surface stiffness on bacterial adhesion. Initially, some of the studies published that the adherence of bacteria onto a surface would increase on a stiffer surface [171,172]. Kolewe’s study demonstrated an increase in bacterial binding on the thinnest PEG hydrogel with a stiffness value of 20 kPa [172]. The authors concluded that the adhered bacteria may be perceived with the stiffness of an underlying glass substrate below the polymer hydrogel. However, different research groups found a decrease in bacterial attachment along with increasing surface stiffness. A study by Song and Ren revealed that more *Pseudomonas aeruginosa* and *Escherichia coli* attachment was observed on the softer surface of cross-linked polydimethylsiloxane (PDMS) [173]. Then, a further study explained that *P. aeruginosa* responded to material stiffness by expressing a high level of intracellular cyclic dimeric guanosine monophosphate (c-di-GMP), a key regulator for biofilm formation [174]. The contradictory observations between Koweli’s study and Song and Ren’s study may be due to the difference in surface hydrophobicity properties, as described in a review [175].

Overall, the mechanical properties of a surface coating are important for long-term stability, durability, and robustness. These properties can be improved by several surface modification methods and be influenced by the material compositions and design of surface coatings. Furthermore, stiffness of surface can also influence the adhesion of bacteria onto the surface, which may enhance antimicrobial activity.

### 3.3. Surface Chemistry Properties

The viability of a virus on any common surface or object is generally because of the strong adsorption process of a virus on inanimate surfaces and objects. The nature of surface materials influences the adsorption of pathogens; as an example, the number of MS2 coliphage plaque was found to be higher on a glass surface than a PVC surface [176]. In addition, the characteristics of bacteria and viruses, such as surface charge, hydrophobicity, size, and shape also play an important role in determining the success of virus adsorption onto the surface [144,177]. Hence, both pathogens and surface must be well characterized and understanding the molecular interaction between adhered pathogens and surfaces is important for fabricating antimicrobial surfaces with appropriate charges or functional groups.

Microbial adsorption on surfaces is mainly regulated by major forces, these are electrostatic interaction (interaction between microbes and surface of opposite charges) and Van der Waals interaction [178]. The charge state of a surface and of microbes affects the microbes’ adsorption rate; when a microbe is attached to an oppositely charged surface, the electrostatic interaction increases. For example, polyglycerol sulfate, which is a polyanion polymer, has a strong binding interaction with herpes simplex virus type-1 (HSV-1), but conjugation with an alkyl chain of fatty amines and nanographene induces a hydrophobic effect that blocks virus fusion and disrupts the membrane [179]. Contrarily, electrostatic repulsion occurs when both virus and surface have similar charges and no adsorption of a virus is observed.

In addition, the net surface charge of a virus is dependent on its isoelectric point (pI) and the pH of surrounding medium. Most viruses have a pI ranging within a pH of 1.9 to 8.4 [180]. At a high pH above the pI of a virus, the net surface charge of the virus becomes negatively charged due to the deprotonation of carboxyl groups found on viral protein. Thus, the adsorption of a virus via electrostatic interaction is favorable on positively charged surfaces such as a polymer with cationic moieties (guanidinium, tertiary sulfonium, primary, secondary, tertiary, and quaternary ammonium) and heterocycle compounds with quaternized nitrogen and quaternized phosphonium groups [181,182]. Meanwhile, in a condition with a low pH below isoelectric point, a virus will have a positive surface net charged due to the excess of hydrogen ions. Interestingly, at isoelectric state, the virus does not carry any surface charge so is not repelling or attracting any surface. Altering the pH of the surroundings may change the behavior of a virus, as described in Nap’s study [183].

Based on the surface charge on a virus, surfaces with cationic moieties or positively charged functional groups are preferable for adsorption of a virus with a negative charge. This is because most viruses are negatively charged on the outer lipid–protein surface. Some cationic polymers were used as functionalized coating for antiviral purposes. Usually, cationic surfaces are designed for contact-killing antiviral action. PEI, made up of repeating units of amine group and a two carbons aliphatic chain, has been used successfully as antiviral coating on a positively charged filter membrane, polyether sulfone (PES) microfiltration membrane, by actively flushing PEI solution through the PES membrane [184]. The coating was effective at showing the virus reduction from drinking water compared to an unmodified membrane due to a combination of the inactivation and adsorption activities of PEI. Unfortunately, the thickness of PEI coating decreased after the filtration process, hence the coating process needs further research. Other than this, a quaternary ammonium compound, a cationic moiety, was covalently bonded onto cellulose, a cationic polymer, to enhance the antimicrobial effect against pathogens [185].

In case of bacterial adhesion, most bacteria possess a net negatively charged cell membrane due to the presence of carboxyl, amino, and phosphate groups on their cell membrane, thus more adhesion was found on positively charged surfaces [186,187,188]. A study by Terada’s group demonstrated that the viability of attached *E. coli* on a positively charged surface was reduced after 8 h of incubation [189]. Positively charged surfaces possess high electrostatic interaction which subsequently induces the loss of bacterial membrane integrity. However, some have reported that a lipopolysaccharide layer on Gram-positive bacteria was able to resist the electrostatic interaction and tightly bind to a negatively charged surface [190]. This study found the the high concentration of c-di-GMP induced the bacteria to change their cell surface when attached to a negatively charged surface.

Moreover, the introduction of hydrophobic moieties on the surface can be used as a strategy to avoid accumulation of microbes on the surface. This is because the presence of a hydrophobic lipid membrane found on viruses will repel a hydrophobic surface, thus preventing the formation of biofilm. Fluorination is one of the processes that creates superhydrophobic surfaces with antimicrobial properties. A study created a low-energy surface by modifying the surface nature using (heptadecafluoro-1,1,2,2-tetrahydrodecyl) trimethoxysilane, known as FAS, creating an interaction between a stainless steel surface and FAS molecules [191]. It was proven that this antimicrobial surface prevented the growth of bacteria on the surface. Furthermore, surface fluorination can also enhance the dispersion stability of nanoparticles in a suspension [192]. Thus, high dispersion stability will avoid particle agglomeration in a coating material.

## 4. Surface Modification Techniques to Improve Surface Properties

Regulating or changing surface properties such as wettability, surface mechanism, and surface chemistry is very essential for providing a high-performance antiviral surface coating. Usually, surface modification is performed for different purposes, such as surface functionalization (introducing new functional groups), surface etching (impurity removal), and surface deposition (deposition of thin layers of coatings). Surface modification techniques can be classified into physical methods and chemical methods. There are various examples of surface modification techniques that can be employed for improving the surface properties of a coating. The choice of modification technique depends on their suitability for intended applications, including their advantages and limitations (Table 3).

Plasma surface treatment is a viable and low-cost surface treatment technique that is effective for most polymeric materials, such as polyvinyl chloride (PVC), polypropylene (PP), and silicone. This treatment involves plasma with positive and negative ions, electrons, and radicals, resulting from existing electric potential difference, reacting, colliding, and breaking covalent bonds available on the targeted surface. Consequently, free radicals will form on the surface and react with oxygen molecules and moisture to produce thermodynamically stable functional groups on the surface (Figure 11) [209]. Therefore, plasma treatment is usually employed to improve surface adhesion coating, remove any foreign contaminants, and modify surface wettability. 

In biomedical application, PP, a synthetic polymer, is usually used for suture material, synthetic grafts, and surgical meshes because of its mechanical strength and safety [210]. However, due to its hydrophobic nature, the PP surface requires surface activation to promote the adhesion to coating materials, wettability, and biocompatibility. PP surface has been functionalized via plasma surface treatment using oxygen gas and argon–oxygen gas [211]. The treated surfaces showed an improvement in surface wetting properties. This is because plasma treatment will chemically induce new functional groups that lower the contact angle of the water droplet on the surface. As previously reported, the new functional groups formed are polar groups or oxygen-containing functional groups such as carbonyl, hydroxyl, and carboxyl groups [212,213,214]. Hence, the changes in the hydrophilicity of a surface may be advantageous for coating adhesion strength for long-term period usage [215].

Morais and his group also demonstrated that plasma treatment can change the surface topography and roughness [211]. Mentioned previously, surface roughness plays an important role in influencing surface wettability and coating adhesion properties. Based on atomic force microscopy (AFM) analysis, both surface roughness values for the treated surfaces via oxygen plasma treatment and argon–oxygen plasma treatment decreased due to plasma physical etching on the surfaces. Two different plasma treatment conditions, determined by the gas used, influence the resultant surface roughness; oxygen plasma treatment and argon–oxygen plasma treatment produced a heterogeneous rough surface and a homogenous, less rough surface, respectively. This observation can be explained by the difference in the atomic radius between oxygen atom and argon atom during plasma physical etching. The relationship between surface roughness and mechanical adhesion strength has been proven in research wherein an increase in surface roughness will promote mechanical interlocking between coating and treated surfaces [216]. It can be assumed that surface roughness will create a large surface area, hence the adhesive becomes stronger.

Apart from using plasma surface treatment, microwave irradiation technique can be used to enhance the surface characteristics of a polymeric surface. Microwaves have electromagnetic waves with a wavelength ranging from 1 mm to 1 m and with frequencies between 300 MHz and 300 GHz. Microwave treatment has been used to modify various surface polymers and fabrics, such as polyurethane (PU), cotton fabrics, and polycarbonate (PC) [195,217,218]. During the process, energy is directly transmitted to the targeted surface through molecular interaction with the electromagnetic field, thus improving surface wettability, roughness, chemical composition, and mechanical properties. A study showed that polypropylene was treated via microwave irradiation in the presence of potassium permanganate as an oxidizing agent [219]. After 120 s of treatment, the results showed that the surface energy of treated polypropylene increased due to the presence of polar components on the surface. An increase in wettability or hydrophilicity has potential to promote a high adhesion strength of coating solution onto PP surface. Similar to a plasma treatment surface, another study used different gases during microwave plasma treatment for different treatment times [217]. Oxygen plasma indicated high surface roughness and a low contact angle compared to argon plasma treatment. Furthermore, the treated surface through the microwave process was able to retain long-term stability of a water contact angle for 100 h at normal conditions [220]. Based on this observation, the water contact angle of the treated surface increases over time but does not reach the initial water contact angle of a pristine surface. Surprisingly, the wettability of this treated surface could be influenced by surface roughness but not by chemical surface composition since there was a slight increase in oxygen to carbon ratio.

Laser surface patterning, also known as laser surface texturing, is one of the surface modification methods wherein a focused laser beam is directed onto a polymeric surface, creating regular or irregular patterns and changing the surface chemistry of the polymer surface [198]. Laser treatment has been used to create superhydrophobic hierarchical structures via femtosecond laser ablation [221]. This laser treatment created dual scale surface structures on treated surfaces which appeared to be strongly hydrophobic or superhydrophobic with contact angle values higher than 140°. Interestingly, the morphology of the ablated surface can be controlled to achieve optimized superhydrophobicity behavior without further post-treatment. This is good for creating a surface with antibiofouling activity. On the other hand, a laser-treated surface can be further treated to impart contact-killing activity on the surface. A study showed that layer-by-layer polyelectrolyte deposition was performed after a laser patterning process led to a driven charge attraction of bacteria [222]. A strong attraction of bacteria onto the surface was observed due to the synergistic effect of a combination of surface opposite charges and micro- and nano-scale structures. Furthermore, laser treatment can assist the deposition of thin layers or nanoparticles onto polymeric surfaces. The irradiation process was carried out in a coating solution wherein laser irradiation onto a substrate surface was likely to induce heterogeneous precipitation of organic molecules, forming a layer coating onto the substrate surface [223]. Similar to Cai et al. (2019), the antimicrobial coating of silver nanoparticles displayed chemical decomposition from micro-drops of silver nitrate solution onto laser-treated polymeric surface [200]. The micro-drops of silver nitrate solution were introduced to the laser ablation zone which was under a high-temperature state. Thereafter, the silver nitrate drops were thermally decomposed into silver nanoparticles and deposited onto the surface. Therefore, laser surface treatment is a time-saving process by cutting out several steps in the synthesis of antimicrobial agents and the coating process.

Apart from this, ultraviolet (UV) irradiation surface treatment has also been used for surface modification. Compared to the other treatments, UV irradiation surface treatment is relatively less expensive, simple, and can operate at a low temperature [224]. The principle of UV surface treatment involves a photosensitized oxidation process which causes the molecules present on pristine polymer surfaces to excite and dissociate due to the absorption of a short wavelength of UV irradiation. Thereafter, reactive sites consist of polar groups, and peroxide functional groups are formed on the surface of pristine polymers. Thus, UV surface treatment is able to tune the surface chemistry and, consequently, surface wettability without changing bulk properties under a low dosage of UV, which are controlled by irradiation treatment time. For instance, UV irradiation treatment has been used to graft a hydrophilic monomer, acrylic acid, onto a low-density polyethylene (LDPE) polymer for food packaging purposes [201]. The introduction of carboxyl groups from acrylic acid enhanced the hydrophilicity of the LDPE surface, which was favorable for the incorporation of antimicrobial agents. The amount of grafted acrylic acid onto the LDPE surface increased as the exposure time was increased but it negatively impacts the mechanical strength of the treated film. Another previous study depicted that prolonging the exposure to UV irradiation caused the formation of a thick silica-like layer on top of the bulk surface of silicone [225]. The formation of a silica-like layer onto bulk poly (dimethyl siloxane) (PDMS) surface acted as a gas diffusion barrier to the bulk surface and maintained its elasticity. In contrast, the formation of a silica-like layer on top of the poly (vinylmethyl siloxane) (PVMS) surface caused substantial changes in the properties for bulk surface due to the susceptibility of vinyl bonds to radical reactions.

Acid or alkaline hydrolysis surface treatment has potential in surface activation and functionalization to improve surface properties. This hydrolysis process involves a chemical bond cleavage due to the nucleophilic attack on atoms available on targeted surfaces, then producing new functional groups. A novel alkaline surface treatment was performed on a titanium surface, producing a porous surface with a hierarchical structure and an open microporosity [164]. In addition, the native titanium oxide layer was attacked by hydroxyl groups from the alkali solution, which subsequently promoted the anchorage of the initiator for the next polymerization process. Another study showed that a combination of acid and alkali treatment on titanium surface will create porous structure inside the surface [226]. Morphological observation proved that acid treatment produced micro-sized pits while alkali treatment produced nano-sized pits that are favorable for protein adsorption onto biomedical implants. The hydrophilicity of the treated surface also improved due to the formation of porous structures. Regarding surface wettability modification, alkaline hydrolysis treatment successfully improved the wettability of the PLA surface in different concentrations of sodium hydroxide [227]. The surface that was treated with a higher concentration of sodium hydroxide had a lower water contact angle than the other. However, prolonging treatment time led to extensive degradation on surfaces in high concentrations of sodium hydroxide, resulting in new rough surface.

Abrasive blasting, also known as sand blasting, is one of the mechanical surface modification techniques used in order to obtain new surface roughness and topography of a substrate. This process involves forcing solid particles or abrasive materials across the hard surface at high-speed using air compression. There are several factors influencing the blasting process, such as the physical properties of grain particles, operation time, working distance between the targeted surface and abrasive materials, and others. A study by Grubova’s research group used a sand blasting pretreatment method on a titanium surface before the coating process [163]. The purpose of this treatment was to roughen the surface of the titanium in order to enhance surface properties such as hardness and strength of coating adhesion. Different sizes of sand particles were used and it was observed that there was a formation of homogenously distributed micro-pits on the titanium surface. The size of the micro-pits formed was linearly dependent on the size of grit particles. Meanwhile, Su and co-workers evaluated the effect of sand blasting conditions on shear bond strength between zirconia (core material in restoration dentistry) and internal resin composite (veneer material) [228]. It was found that as the pressure applied increased from 0.2 to 0.6 MPa during the process, the shear bond strength was significantly higher at 0.1 MPa. Under high pressure, several surface properties such as roughness, bonding area, and wetting behavior of adhesives were enhanced. However, the probability of surface defects and flaws can increase as the pressure increases, subsequently negatively affecting the surface bonding. In addition, surface roughness will increase as particle size and sand blasting time increase. Thus, several blasting parameters need to be considered to achieve a well-defined surface.

Furthermore, some modification techniques are not limited to modifying the surface properties but can also be used for coating deposition. As an example, a chemical vapor deposition polymerization method involves vapor-phase monomers of a coating that form a solid polymeric film deposited onto a substrate surface. The deposition of organic coating onto a metal surface has been successfully achieved via chemical vapor deposition method [229]. Different types of calcium phosphate forming on the metal substrate could be controlled by changing the precursor temperature, eventually affecting the evaporation rate and molar ratio of the precursors. The substrate surface roughness also influenced the resulting morphology and thickness of coating. Initiated chemical vapor deposition, a new chemical vapor phase method, has been introduced. The key characteristic of initiated chemical deposition is the introduction of an initiator species that induces vapor phase monomer deposition at a high rate under mild conditions [206]. Hence, it is suitable for heat-sensitive and fragile polymer substrates. In a study, nylon fabric was coated with poly(dimethylaminomethyl styrene) coating by using initiated chemical vapor deposition [230]. The coating method did not affect the color or feel of the fabric as a result of no coating occluding the pores inside fabric. In addition, the antimicrobial coating was strongly coated onto the fabric because of an insufficient amount of antimicrobial agent present in the supernatant from a fabric shake test to inhibit bacterial growth.

Moreover, click grafting coating has been used to modify an inert polymeric surface membrane for biomedical purposes [231]. This developed coating has been successful for improving blood compatibility with less than 2% of hemolytic index by introducing a chitosan biopolymer. This simple surface modification process involves several steps; these are amination, activation, and grafting steps which increase surface roughness of native polyvinyl chloride due to the substitution of chlorine groups with primary amine groups, ethylene diamine (EDA). An increase in surface roughness will enhance the adhesion capability between PVC and chitosan with the presence of glutaraldehyde. Mandolfino et al. (2014) agreed that surface roughness enhanced the mechanical adhesion strength because roughness will promote mechanical interlocking between the coating and the treated surfaces [216]. In addition, the presence of functional groups significantly affects the network elasticity of PVC, which is reflected in an increase in maximum stress and a decrease in maximum strain.

Overall, all surface modification techniques listed are capable of modifying and enhancing surface properties that might be useful for fabricating an effective antiviral surface coating. Most of the techniques will affect surface roughness, surface chemistry composition, and surface wetting properties, which lead to the improvement in other surface properties, such as adhesion strength. 

## 5. Current Antimicrobial Coating to Combat COVID-19

Due to the rising number of COVID-19 cases around the world, many efforts have been made and resources used in attempting to stop or combat the transmission of this deadly new coronavirus, SARS-CoV-2. Currently, antimicrobial surface coatings are being developed and fabricated by several research groups to be applicable to various surfaces, thus reducing the risks of environmental microbial infection. For instance, Mantlo and co-workers evaluated the effectiveness of an established antimicrobial surface coating, Luminore CopperTouch surface coating, against multiple viruses, including SARS-CoV-2 [232]. The targeted surfaces were sprayed with these copper and copper–nickel based antimicrobial surface coatings after the disinfection step. It was found that the copper sprayed surface was able to inactivate 99% of SARS-CoV-2 virus after 2 h of exposure, while both copper and copper–nickel sprayed surfaces could inactivate 99% of Ebola and Marbug viruses after 30 min of exposure. The difference in inactivation time between the Ebola virus and SARS-CoV-2 virus can be related to the structure of the tested viruses which affects virus susceptibility [233]. In comparison, the Ebola virus has a long, filamentous shape, whereas SARS-CoV-2 is spherical with spike proteins. Therefore, the Ebola virus has more surface area in contact with the surface, while SARS-CoV-2 has a distant length between viral capsid and copper surface.

Next, another study successfully developed a new antimicrobial film coating wherein metal oxide was incorporated into a polyurethane-based coating [152]. This antimicrobial film was developed via a curing process at 120 °C onto the surface of substrates (glass and stainless steel). After the curing process, the treated substrates were plasma-cleaned to remove excess polyurethane from the substrate. Based on the conducted antiviral tests, this coating was able to reduce SARS-CoV-2 viral titer by about 99.9% after 1 h of exposure. The antimicrobial activity shown by cuprous oxide may involve the dissolution of copper ions, the production of ROS species, direct contact with cuprous oxide, and viral entry inhibition into host cells. Interestingly, the required virus inactivation time was influenced by surface coating wettability in which the surface with the lower contact angle had a greater contact area with viral suspension. As a result, faster penetration of viral liquid suspension into the antimicrobial film coating could occur and, hence, less time course for virus inactivation was required. As evidenced, the hydrophobic coating took about 24 h to inactivate 99.9% of SARS-CoV-2 virus, while the hydrophilic coating only required an hour for inactivation.

Nevertheless, hydrophobic surface coating can also act as an antibiofouling coating that repels any microbial adhesion on the surfaces. Nie and colleagues fabricated a superhydrophobic silane-based surface coating with multiple properties such as antimicrobial and anticorrosion abilities with enhanced wear resistance [170]. A ‘sandwich’-like structure was designed and composed of a chemically etched metal surface at the bottom, a silicon dioxide nanoparticle-hybridized silane layer in the middle, and a carbide nanosheet-hybridized silane layer on the top of the surface. Each of the coating layers were spun-coated onto the pretreated surface for 30 s at 3000 rpm. As a result, the introduction of a silane layer as the film caused a high contact angle (exceed 160°) which made water droplets roll off the surface easily. In addition, high bacteriostatic efficiency was observed because of the presence of carbide nanosheets in the coating. So far, this research group suggested that this coating strategy has great potential for being used in combating COVID-19. However, no further antiviral activity against the SARS-CoV-2 virus has been evaluated.

Furthermore, the usage of quercetin-based coating could be a good candidate as an antimicrobial film. In the past, quercetin flavonoid compound has only been used as a therapeutic agent for various diseases such as cancer, diabetes, cardiovascular diseases, and bacterial infections and viral infections [234]. Unfortunately, there are not so many studies on quercetin-based surface coating. Recently, Cristescue et al. evaluated the potential of quercetin as an antimicrobial compound incorporated in coating film [235]. In their attempt, quercetin compound was embedded into polyvinylpyrrolidone (PVP) biopolymer and deposited onto a glass slide by using matrix-assisted pulsed laser evaporation (MAPLE) technology. By using this deposition technique, the obtained film was thin and had good surface uniformity on the glass surface. Then, an antibiofilm assay proved that this coating film had significant antibiofilm properties against Gram-positive and Gram-negative bacterial strains within 72 h. Therefore, the study by Cristescue proposed that this flavonoid-containing coating could be directed to eradicate COVID-19 transmission. This is because quercetin, a natural flavonoid product, can act as an inhibitor of SARS-CoV-2 protease, which is responsible for the viral replication cycle [236].

During this pandemic outbreak, some researchers have been concerned about the microbial food safety of various food products and the tendency of COVID-19 transmission from food packaging to consumers, including food contaminated with the virus [237]. Hence, developing a packaging material with an antimicrobial coating can be a great solution to avoid this problem. Mizielinska and co-workers successfully produced polyethylene packaging with an antimicrobial coating based on zinc oxide nanoparticles [238]. The metal nanoparticle solution was introduced into a polymer matrix via a sonication process until a uniform mixture solution formed. Based on antiviral analysis, coating with embedded zinc oxide nanoparticles demonstrated a log reduction in viral titer after 16 h of analysis. Interestingly, the usage of zinc oxide nanoparticles in the external coating of food packaging could provide shielding properties against UV light to maintain the antimicrobial properties of the coating. In addition, this coating containing metal nanoparticles was also supplemented with additional antimicrobial agents, named geraniol and carvacrol, separately. It was observed that the addition of zinc nanoparticle into the coating, based on geraniol and carvacrol, led to an improvement in antiviral activity. However, the demonstrated antiviral activity was moderate due to only reducing viral titer, devoid of the complete inactivation of virus particles.

Furthermore, face masks have been recommended as protection tools against COVID-19 and the excessive demand for them can pose significant challenges to the environment. There are also possibilities of viral transmission occuring after touching a contaminated face mask, and some viral loads can escape from the masks. Thus, some research studies are focusing on fabricating a face mask with an antimicrobial coating based on graphene, metal nanoparticles, or quaternary ammonium compounds. In a study by Zhong et. al., a self-disinfecting face mask was developed with superhydrophobic coating onto N95 respirators [154]. This superhydrophobic coating was composed of polyimide film with the presence of silver nanoparticles and laser-induced graphene by using laser-induced forward transfer process. As a result, this respirator has the potential to inactivate SARS-CoV-2 via the synergistic effect of superhydrophobic coating with plasmonic heating properties alongside with silver ion release towards microbes. The presence of silver nanoparticles in the coating plays an important role in the plasmonic effect, which can raise the surface temperature by up to 80 °C within 1 min of sunlight illumination. In addition, the introduction of graphene could render the surface of an N95 respirator as superhydrophobic with a contact angle over 140°. However, this study requires further attention on the mechanical stability of this N95 respirator by adding polymer materials.

Another study also demonstrated the efficiency of a functionalized graphene-based filter for bacterial filtration efficiency and viral inactivation [239]. The functionalized graphene-based filter on a 3D-printed face mask showed the highest bacterial filtration efficiency (98.02%) with a lower breathing resistance value in comparison with a commercialized face mask. This functionalized graphene filter was found to completely arrest the transmission of SARS-CoV-2 viral particles. Likewise, another antimicrobial face mask was successfully developed with incorporated benzalkonium chloride via dip coating method [240]. This non-woven face mask was capable of inactivating more than 99.9% of SARS-CoV-2 virus within one minute of contact due to the antimicrobial activity of benzalkonium chloride, a quaternary ammonium compound. Moreover, a thin antimicrobial composite coating based on silver nanocluster/silica was sputter-coated directly onto disposable facial masks [128]. This deposition method produced a thin coating of less than 200 nm, and the amount of silver nanocluster was controlled by changing the power value in direct current (DC). As a result, face masks coated with a high amount of silver nanocluster can completely reduce the titer of SARS-CoV-2. Overall, all of the antimicrobial coatings produced could be adapted for the production of other antimicrobial clothes, gloves, and common surfaces.

## 6. Summary and Future Direction

In response to the COVID-19 outbreak, the development of an antimicrobial surface coating is one of the best preventive solutions to mitigate the spread of this SARS-CoV-2 virus. Until now, there have been many attempts to fabricate antimicrobial surface coatings. Various types of antimicrobial agents such as antimicrobial polymers, metal-based antimicrobial agents, and antimicrobial nanoparticles have been developed as replacements to conventional antibiotic compounds. Various antimicrobial mechanisms have been employed by these antimicrobial agents in order to inhibit the growth of pathogens and viruses. Sometimes, the usage of two or more antimicrobial agents can enhance antimicrobial activity. Furthermore, several essential surface properties, such as surface wettability, surface mechanical properties, and surface chemistry properties, are important and should be considered during the fabrication of an antimicrobial coating to make it more durable and long-lasting without decreasing the antimicrobial activity of antimicrobial materials or agents. In terms of the wetting surface property, a hydrophobic surface may be good for preventing the adsorption of virus-contaminated water molecules or biofilm formation. A hydrophilic surface can also be used in order to design a multilayer coating so that the adhesion of the layers is strong enough. The mechanical adhesion strength of a coating onto a polymer substrate is dependent on wettability and surface roughness. Chemical surface properties, such as surface charge, and the functional groups present on the surface may influence the adsorption of a virus due to the interactions that arise between virus and surface. Most of the cationic surface is favorable for virus and bacterial adsorption. Following that, there are some surface modification techniques that can be used to improve these surface properties. Usually, the aim of performing surface modification is to improve the hydrophilicity of a surface with low energy and eventually assist in coating deposition. In future, improvement in these developed antimicrobial surface coatings can be achieved, such as by developing a coating that is responsive or sensitive to stimuli. Other than that, the biocompatibility of these antimicrobial coatings to humans can be investigated in order to make it easy to apply to existing common surfaces, such as stationaries, keys, or doorknobs. Overall, an antimicrobial surface coating has great potential for curbing this current outbreak or any upcoming pandemic in future.

## Figures and Tables

**Figure 1 polymers-13-04234-f001:**
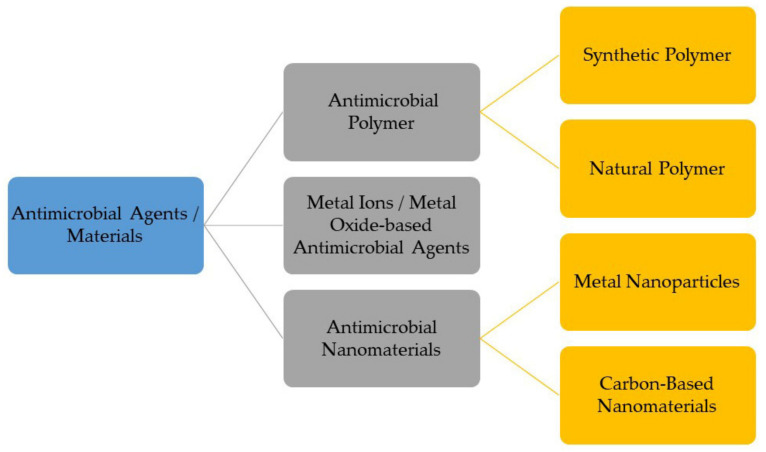
Classification of various antimicrobial agents/materials for development of antimicrobial surface coating.

**Figure 2 polymers-13-04234-f002:**
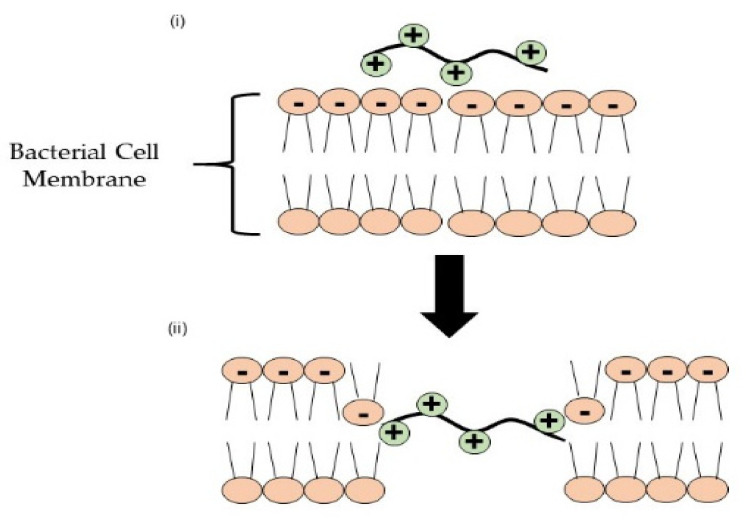
Antimicrobial mechanism of cationic polymer on bacterial cell membrane. (**i**) Adsorption of cationic polymer onto bacterial cell membrane via electrostatic interaction and (**ii**) insertion of cationic polymer into phospholipid membrane bilayer causing translocation of anionic lipids and leading to cell burst.

**Figure 3 polymers-13-04234-f003:**
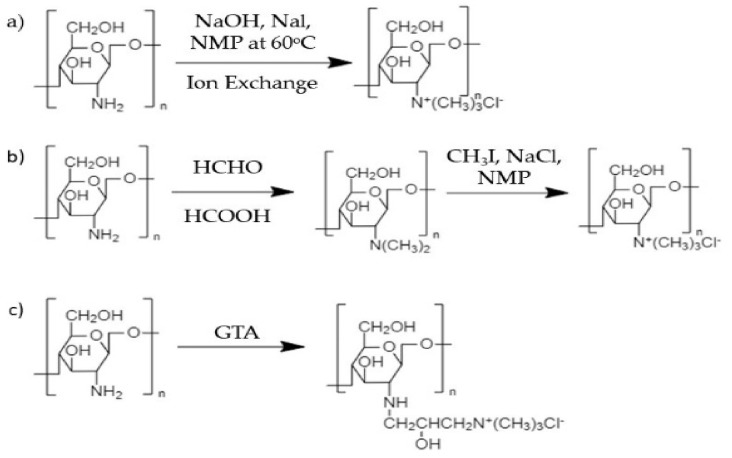
Several synthesis reactions for quaternized chitosan derivatives: (**a**) Direct quatenization of chitosan producing TMC, (**b**) N-alkylation of TMC, and (**c**) epoxy-derivative ring-opening producing *N*-((2-hydroxy-3-trimethylammonium)propyl) chitosan chloride (HTCC).

**Figure 4 polymers-13-04234-f004:**
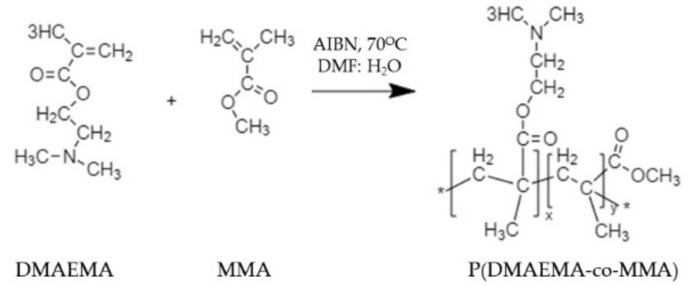
Synthesis of P(DMAEMA-co-MMA), an amphiphilic copolymer from hydrophilic DMMAEMA monomer and hydrophobic MMA monomer.

**Figure 5 polymers-13-04234-f005:**
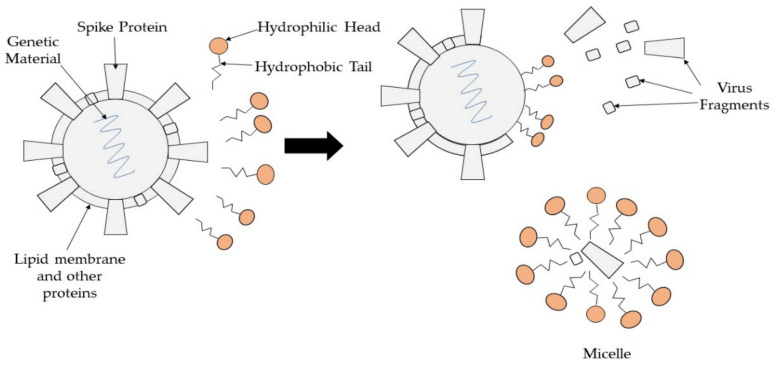
Mechanism of surfactants for inactivating virus.

**Figure 6 polymers-13-04234-f006:**
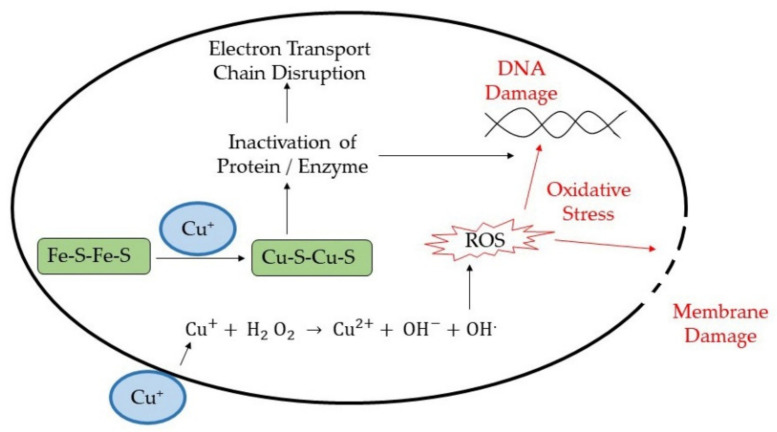
Antimicrobial mechanism of copper ions through ROS production and metal donor atom selectivity.

**Figure 7 polymers-13-04234-f007:**
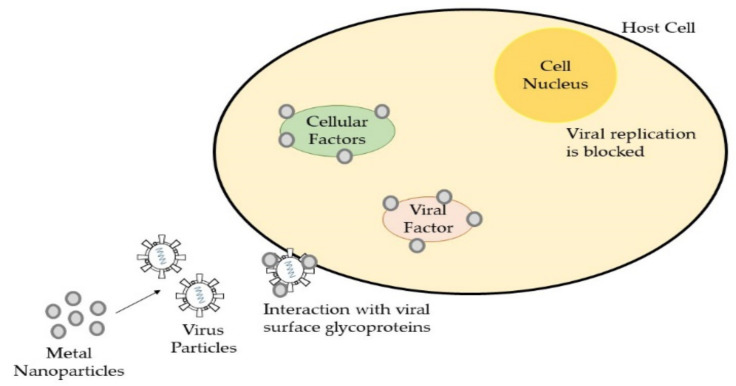
Antiviral mechanism of metal nanoparticles during virus infection.

**Figure 8 polymers-13-04234-f008:**
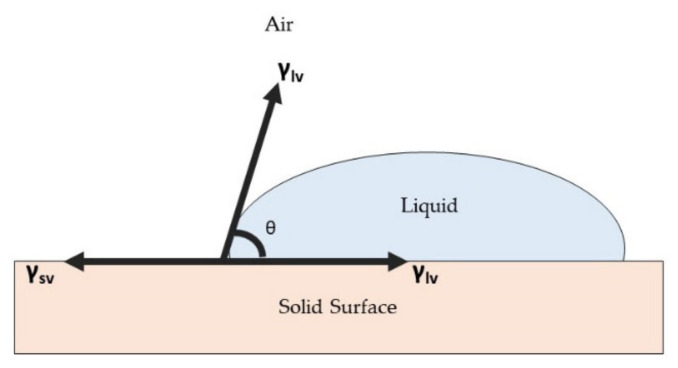
Three different interfacial boundaries’ contact line for water contact angle.

**Figure 9 polymers-13-04234-f009:**
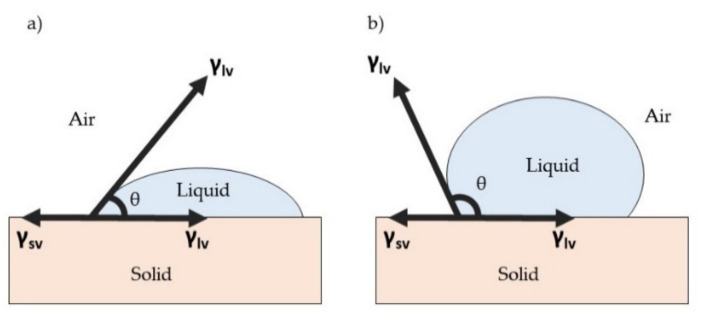
Contact angle of the liquid droplet between (**a**) hydrophilic surface and (**b**) hydrophobic surface.

**Figure 10 polymers-13-04234-f010:**
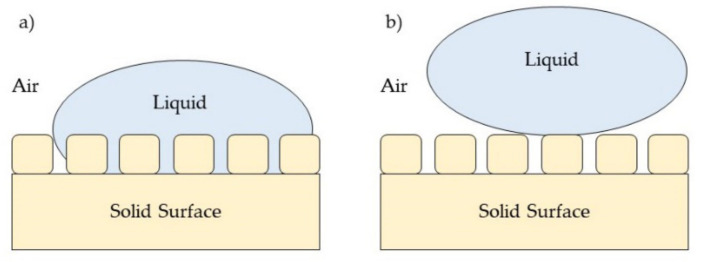
Wetting of liquid droplets on rough surfaces. (**a**) Wenzel model and (**b**) Cassie–Baxter model.

**Figure 11 polymers-13-04234-f011:**
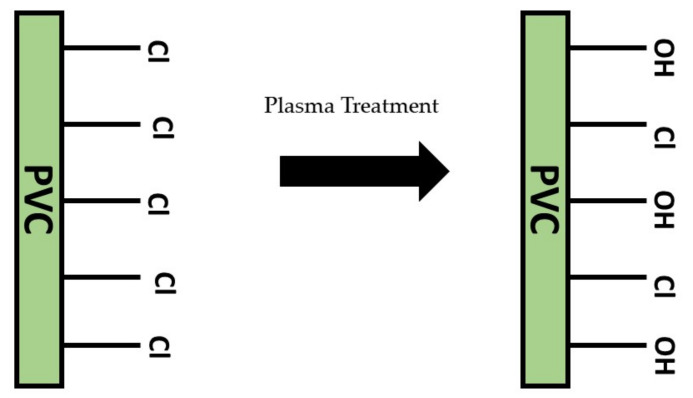
Plasma surface treatment changing surface morphology and functional group formation on the treated surface.

**Table 1 polymers-13-04234-t001:** Application of antimicrobial polymer coating with their antimicrobial activity efficiency.

Coating Materials	Coating Techniques	Microbes	Antimicrobial Activity	Application	Ref.
**Synthetic Coating Materials**					
Poly(allylamine)-poly(sodium 4-styrenesulfonate)/poly(allylamine hydrochloride) multilayer	Plasma polymerization and layer-by-layer assembly	*Staphylococcus aureus* and *Escherichia coli*	77.78 ± 1.72% 95.15 ± 2.40% The antibacterial capability is measured based on antibacterial ratio	Titanium implant antibacterial coating	[27]
An amphiphilic polymer made up of polyoxypropylene (poly(propylene oxide)) flanked with two hydrophilic chains of polyoxyethylene (poly(ethylene oxide)), embedded with chlorine dioxide, copper, and L-vitamin C	Not mentioned	Influenza A (H1N1), methicillin-resistant *Staphylococcus aureus* (MRSA) and *Acinebacter baumannii*	No virus plaque observed After 1 min of contact, viral protein envelope is damaged Over 99.9% of antimicrobial activity after 5 min of contact time	Antipathogenic coating for additional measure	[28]
PEI-silver nanoparticle and copper nanoparticle membrane	Covalent linking via layer-by-layer	MS2 bacteriophage	4.5 to 5 log reduction	Membrane filter for drinking water	[29]
Polyvinylpyyrolidone/titanium dioxide	Simple dip coating	*Escherichia coli*	The width of inhibition zone ranges from 4.5 to 8 mm	Medical device coating with improved blood compatibility and antimicrobial activity	[30]
*N*,*N*-dodecyl, methyl-PEI	Physical painting using cotton swab	Influenza A wild-type and resistant type (H3N2) and avian influenza A wild-type and resistant type (H4N2)	100% biocidal efficiency for all tested viruses	Antiviral surface painting	[31]
Poly(hydantoinylacrylamide-co-3-(trimethyoxysilyl)propyl methacrylate) (HASL)	Covalent binding with cellulose cotton fabric	*Staphylococcus aureus* and *Escherichia coli*	About 6 log reduction in all tested microbes	Not mentioned	[32]
Poly(hydantoinylacrylamide-co-glycidyl methacrylate) (HAGM)	Covalent binding with cellulose cotton fabric				
Poly(hydantoinylacrylamide-co-2-hydroxyethyl methacrylate) (HAOH)	Cross-linking via an agent to cellulose cotton fabric				
Poly(L-lactide)/poly (ε-caprolactone)/propolis	Solvent casting	*Staphylococcus aureus*	Inhibition zone diameter ranges from 13 to 17 nm	Guided tissue regeneration application	[33]
*N,N*-hexyl, methyl-PEI	Covalent attachment	Poliovirus	100% virucidal activity	Aqueous solution disinfection	[34]
*N,N*-dodecyl, methyl-PEI	Physical painting	Influenza virus, *Staphylococcus aureus,* and *Escherichia coli*	100% virucidal and bactericidal activity	Not mentioned	[35]
Polyester/polyurethane/levofloxacin	Hot-press polymer immobilization	*Staphylococcus aureus*	No viable bacteria found on coated substrate	Antimicrobial implant coating application	[36]
**Natural Coating Materials**					
Carboxymethylcellulose/chitosan multilayer	Chemical cross-linked layer-by-layer assembly	*Staphylococcus aureus* and *Pseudomonas aeruginosa*	74% reduction at 24 h 83% reduction at 72 h	Superhydrophilic coating for ophthalmic applications	[37]
Carrageenan/green tea extract	Simple dip coating	Murine norovirus (MNV-1) and hepatitis A virus (HAV)	Below detection limit at any condition Lower than 3 log reduction	Antiviral edible coating for fruits	[38]
Chitosan/green tea extract film coating	Solution casting onto polypropylene film	Murine norovirus (MNV-1)	1.6 to 4.5 logs PFU/mL reduction after 24 h incubation	Active food packaging	[39]
Chitosan	Covalent linking via silanization step	*Escherichia coli* and *Staphylococcus aureus*	No viable cells observed after 24 h	Antibacterial surface for biomedical devices	[40]
Carrageenan/citric acid	Not mentioned	*Staphylococcus aureus, Dickeya chrysanthemi, Escherichia coli, Proteus mirabilis,* and *Pseudomonas aeruginosa*	Inhibition zone diameter for carrageenan film with highest concentration of citric acid ranges from 3.25 ± 0.29 mm to 4.18 ± 0.28 mm	Biodegradable film	[41]
Gelatin/chitosan/d-limonene	Solvent casting	*Escherichia coli*	Film containing highest d-limonene concentration has inhibition zone diameter with 22.0 ± 1.2 mm	Antimicrobial edible film for food packaging	[42]
Polyelectrolyte multilayer composed of carrageenan and chitosan embedded with nisin Z	Layer-by-layer coating	*Staphylococcus aureus* and MRSA	Kill over 90% and 99% of planktonic and biofilm cells, respectively	Antimicrobial multilayer coating	[43]

**Table 2 polymers-13-04234-t002:** Inorganic metal nanoparticles: sources, physical characteristics, and antimicrobial activity.

Inorganic Metal Nanoparticle	Synthesis Route	Size and Shape	Antimicrobial Activity	Ref.
Ag	Biological synthesis using *Olea europea* aqueous extract	11.6–20.7 nm Spherical	Inhibition zone diameter ranged from 14 to 22 mm for *Streptococcus mutans* and 7 to 13 mm for *Candida albicans*	[113]
Ag	Biological synthesis using *Lamptanthus coccineus* aqueous and hexane extract *Malephora lutea* aqueous and hexane extracts	10.12–27.89 nm Spherical 8.91–14.48 nm Spherical	The antiviral activity was measured based on IC_50_ ^1^ (µg/mL) For HAV-10 virus: no activity for aqueous extract 11.71 for hexane extract For HSV-1 virus: 520.6 for aqueous extract 36.36 for hexane extract For CoxB4 virus: no activity for aqueous extract 12.74 for hexane extract For HAV-10 virus: no activity for aqueous extract 31.38 for hexane extract For HSV-1 virus: no activity for both aqueous and hexane extract For CoxB4 virus: 46.44 for aqueous extract 29.04 for hexane extract	[114]
Ag	Biological synthesis using bacterial enzyme	77–92 nm Spherical, triangular, and hexagonal	Inhibited the growth of Bean Yellow Mosaic Virus	[115]
Silver oxide (AgO)	Biological synthesis using bioactive compounds from *Oscillatoria* sp.	14.42–48.97 nm Spherical	49.23% reduction of HSV-1 reproduction in dilution ranging from 10^−1^–10^−8^	[116]
Au	Biological synthesis using bioactive compounds from *Spirulina platensis*	15.60–77.13 Octahedral, pentagonal, and triangular	42.75% reduction of HSV-1 reproduction in dilution ranging from 10^−1^–10^−8^	
Copper oxide (CuO)	Biological reduction using *Momordica charantia* aqueous extract	61.48 ± 2 nm Rod-shaped	Inhibited *Bacillus cereus* with 31.66 nm zone of inhibition 80% viability of infected embryo with Newcastle Disease Virus (NDV) was observed by using 100 µg/mL concentration of CuO nanoparticles	[117]
Manganese (Mn)	Biological reduction using curcumin ethanolic extract	In the range of 50 nm Spherical	Inhibition zone diameter ranged from 11 to 20 mm for various bacterial species and fungal species	[118]
Ag	Biological reduction using *Citrus limetta* peels	5 nm Spherical	More than 90% inhibition against chikungunya virus (CHIKV) at different nanoparticle concentrations (0.05 mg/mL, 0.1 mg/mL, and 0.2 mg/mL)	[119]
Iron (Fe)		32 nm Spherical		
ZnO		12 nm Spherical		
Aluminum oxide (Al_2_O_3_)	Biological reduction using *Cymbopogan citratus* leaf extract	34.5 nm Spherical	Complete growth inhibition against *Pseudomona aeruginosa*	[120]
Titanium dioxide (TiO_2_)	Biological reduction using *Psidium guajava* leaf extract	32.58 nm Spherical	Maximum inhibition zone diameters achieved were 25 mm and 23 mm for *Staphylococcus aureus* and *Escherichia coli*, respectively	[121]
Nickel oxide (NiO)	Biological reduction using *Eucalyptus globulus* plant extract	10 to 20 nm	Inhibition zone diameter ranged from 13 to 17 mm for various bacteria	[122]
Au-Ag-zinc ZnO-chlorine dioxide nanocomposite	Chemical reduction using citric acid	20–40 nm for AuNP 10–40 nm for AgNP 25–35 nm for ZnO Nanoparticle	Inhibited 93.5–100% of SARS-CoV-2 formation	[123]
Au	Chemical reduction using mixture of tetraethoxysilane and triethoxysilane	1.5–20 nm Spherical	55–96% inhibition of adenovirus reproduction in MDBK cell culture at various nanoparticle dilutions	[124]
ZnO	Chemical synthesis via molten salt method	39.7 nm Star-like shape	Growth curve of both *Bacillus subtilis* and *Enterobacter aerogenes* decreased after 24 h of incubation	[125]
CuO	Chemical synthesis using sodium hydroxide	Average diameter is 10 nm Nanorod	99%, 98%, and 93% growth reduction in *Escherichia coli, Shigella flexneri*, and *Staphylococcus aureus*, respectively	[126]
Ag	Electrochemical	7.1 nm Quasi-spherical	Effective concentration was 3.13 ppm against poliovirus	[127]
Ag nanocluster with silica composite	Radio frequency co-sputtering process with argon	Less than 200 nm	100% inhibition against coronavirus	[128]

^1^ IC_50_ is the half maximal inhibitory concentration of antimicrobial agent and is measured in µg/mL.

**Table 3 polymers-13-04234-t003:** Advantages and limitations of surface modification techniques.

Modification Techniques	Advantages	Disadvantages	Ref.
Plasma Surface Treatment	Low-cost, reliable, and reproducible, Short surface treatment time, Versatility, can be used for diverse range of surface materials, Environmentally friendly and operator friendly, Can be scaled up to industrial production.	High investment cost, Effective plasma dose determination without damaging treated substrate, Decontamination of uneven surfaces may be inefficient.	[193,194]
Microwave Radiation	Non-contact heating, suitable for heat-sensitive materials, Relatively low cost, energy, and treatment time compared to others, Good instantaneous control and reduced environment pollution.	Low productivity.	[195,196,197]
Laser Surface Texturing or Patterning	Can modify polymeric surfaces at a macro-, micro-, and nano-size scale with a high spatial and temporal resolution, Contamination can be easily avoided due to non-contact treatment, High processing speed, high automation, and possibility to treat large areas, No utilization of harmful chemical reagents.	Costly.	[198,199,200]
Ultraviolet Irradiation Surface Treatment	Fast reaction rate, Low cost of processing, Relatively simple process equipment.	Non-uniform and low density of surface functionalization.	[193,201,202]
Acid/Alkali Hydrolysis	Increased surface energy, Removes contamination, Low cost and simple process, High selectivity.	The introduction of oxygen containing functional groups onto the surface is non-specific, Difficult to be scaled up, Residual ion deposition.	[193,203,204]
Abrasive Blasting or Sand Blasting	Produce uniform roughness on the surface, Can be applied onto surfaces with irregular shape.	Possibility for leaving contaminants on the treated surface.	[199,205]
Chemical Vapor Deposition	Solvent-free process, Producing a highly uniform coating on complex geometries.	Requires highly specialized equipment, High initial investment.	[203,206]
Click Grafting	Easy introduction, Controllable density, Exact localization of graft chains at the surface without changing bulk properties of substrate.	Requires additional processing steps.	[207,208]

## Data Availability

Not applicable.

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
