# Peer review of "Past and Current Progress in the Development of Antiviral/Antimicrobial Polymer Coating towards COVID-19 Prevention: A Review"

_polymers, 2021, doi:10.3390/polym13234234_

Round 1

Reviewer 1 Report

Well written, actual and coincise. 

Author Response

Dear Reviewer #1,

Many thanks for your kind feedback. We are highly appreciated the time taken to read and review the submitted review paper. We believed the review paper has a significant impact on the reader of the research field and hope that it will be one of the referred papers for the antibacterial/antiviral coating works. Thank you again for your comment.

Reviewer 2 Report

This article was written by seven authors. Their aim was to provide an overview on various types of antiviral or antimicrobial coating agents such as antimicrobial polymer-based coating, metal-based coating, functional nanomaterial and nanocomposite-based coating. The action mode for each types of antimicrobial agents against pathogens are elaborated. In addition, surface properties of the designed antiviral and antimicrobial polymer coating with theirs influencing factors are discussed in this review. Chemical surface properties such as surface charge, wettability, surface roughness and functional groups present on the surface This paper also exhibited several techniques on surface modification to improve surface properties.

Comments and suggestions for authors:

The article is well-organized. It gives an appropriate overview and summary of this very topical area of research. The authors did a good job of synthesizing summarizing 242 references. The article well-written and easy to understand. It contains very few spelling errors. Table 3 contains structural errors that can be easily fixed.

Author Response

Dear Reviewer #2,

Many thanks for the summarized comments on the content of the submitted review manuscript. We are grateful to have the opportunity to write a comprehensive topical review on the antibacterial/antiviral coating. Since the outbreaks of Covid-19 virus, the need to specific antiviral coating is elevated. Thus, it is significant to disseminate the principles of antibacterial/antiviral coating to help the researchers in the field of research to develop coating for antiCovid applications. We believed the review paper has a significant impact on the reader of the research field and hope that it will be one of the referred papers for the antibacterial/antiviral coating works especially for antiCovid coatings. Thank you again for your comment

Dear Reviewer #2,

Thank you for the time taken to read and reviewed the submitted review manuscript. We had amended the structural errors for Table 3 as requested by the reviewer #2. The typo errors were also addressed. We believed the review paper has a significant impact on the reader of the research field and hope that it will be one of the referred papers for the antibacterial/antiviral coating works especially on antiCovid coatings. Thank you again for your comment.
